

# Elucidating the role of nicotinamide N-methyltransferase-p53 axis in the progression of chronic kidney disease

Xin Zhen[1], Yuxiang Sun[1], Hongchun Lin[1], Yuebo Huang[1], Tianwei Liu[2], Yuanqing Li[1] and Hui Peng[1]

[1] Nephrology Division, Department of Medicine, The Third Affiliated Hospital of Sun Yat-sen University, Guangzhou, Guangdong, China
[2] Zhongshan School of Medicine, Sun Yat-sen University, Guangzhou, Guangdong, China

Corresponding author
Hui Peng, pengh@mail.sysu.edu.cn

## ABSTRACT

**Background:** Chronic kidney disease (CKD) is a significant global health issue characterized by progressive loss of kidney function. Renal interstitial fibrosis (TIF) is a common feature of CKD, but current treatments are seldom effective in reversing TIF. Nicotinamide N-methyltransferase (NNMT) has been found to increase in kidneys with TIF, but its role in renal fibrosis is unclear.
**Methods:** Using mice with unilateral ureteral obstruction (UUO) and cultured renal interstitial fibroblast cells (NRK-49F) stimulated with transforming growth factor-β1 (TGF-β1), we investigated the function of NNMT *in vivo* and *in vitro*.
**Results:** We performed single-cell transcriptome sequencing (scRNA-seq) on the kidneys of mice and found that NNMT increased mainly in fibroblasts of UUO mice compared to sham mice. Additionally, NNMT was positively correlated with the expression of renal fibrosis-related genes after UUO injury. Knocking down NNMT expression reduced fibroblast activation and was accompanied by an increase in DNA methylation of p53 and a decrease in its phosphorylation.
**Conclusions:** Our findings suggest that chronic kidney injury leads to an accumulation of NNMT, which might decrease p53 methylation, and increase the expression and activity of p53. We propose that NNMT promotes fibroblast activation and renal fibrosis, making NNMT a novel target for preventing and treating renal fibrosis.

## INTRODUCTION

Chronic kidney disease (CKD) is a leading cause of death worldwide, with its morbidity and mortality increasing annually (*Webster et al., 2017*). The progression of CKD to end-stage renal disease (ESRD) is often accompanied by tubulointerstitial fibrosis (TIF), characterized by the excessive deposition of extracellular matrix and ultimately leading to renal function loss. The activation and proliferation of renal fibroblasts are key events in the TIF process (*Kuppe et al., 2021*). After injury, renal tubular epithelial cells secrete cytokines such as transforming growth factor-β1 (TGF-β1), connective tissue growth

factor (CTGF), and platelet-derived growth factor (PDGF), which promote the activation of myofibroblasts and the production of extracellular matrix (ECM) (*Edeling et al., 2016*). Excessive ECM deposition increases kidney stiffness, further impeding the perfusion and filtration functions of the glomerulus. Several signaling pathways, including TGF-β/Smad, signal transducer and activator of transcription 3 (STAT3), and p53, have been implicated in the activation of renal myofibroblasts (*Liu, 2011*).

Nicotinamide N-methyltransferase (NNMT) is a cytosolic enzyme that uses the universal methyl donor S-adenosyl methionine (SAM) to convert nicotinamide (NAM) to N1-methylnicotinamide (MNAM), producing S-adenosyl-L-homocysteine (SAH) in the process (*Kraus et al., 2014*). NNMT plays a significant role in regulating methyl donor balance and is associated with DNA and histone methylation, involved in multiple metabolic pathways in both normal tissues and cancer cells (*Crujeiras et al., 2018*). Prior research has suggested that the NNMT metabolite MNAM may alleviate lipotoxicity-induced kidney tubular injury and tubulointerstitial fibrosis (TIF) (*Tanaka et al., 2015*; *Zhang et al., 2022*). Although NNMT is primarily expressed in the liver and adipose tissue, low levels of expression have also been detected in healthy kidneys, hearts, and brains (*Lu et al., 2022*). Additionally, a previous study found that NNMT is highly expressed in the matrix surrounding cancer cells and is considered a key regulatory molecule in maintaining fibroblast phenotype (*Eckert et al., 2019*). However, the precise biological roles of NNMT in renal fibrosis remain unclear.

The classic tumor suppressor, p53, plays a pivotal role in cellular signaling by contributing to DNA repair and modulating cell cycle arrest following DNA damage or cellular senescence (*Levine, 2020*; *Mijit et al., 2020*). Studies have shown that p53, working as a crucial cofactor in TGF-β1-mediated transcription of profibrotic genes, is activated in kidney extracellular matrix remodeling *via* a TGF-β-dependent nonclassical pathway (*Overstreet et al., 2014*). The activation of p53 is regulated by post-translational modifications, such as phosphorylation, acetylation, and methylation (*Chmelarova et al., 2013*). CpG island hypermethylation in the p53 promoter region can downregulate its expression and prevent transcription factor binding (*Ghazi, Nagiah & Chuturgoon, 2021*). Besides DNA methylation of CpG islands, p53 is one of the few transcription factors that undergo protein methylation regulation (*Scoumanne & Chen, 2008*). Methyltransferases facilitate p53 protein methylation by using SAM as a methyl donor, producing S-adenosyl-L-homocysteine that is further degraded into adenosine and homocysteine (*Schubert, Blumenthal & Cheng, 2003*).

In this study, we discovered that NNMT was highly expressed in the kidneys of mice with unilateral ureteral obstruction (UUO). Our scRNA-seq results indicated that NNMT was predominantly expressed in renal fibroblasts after UUO injury. We demonstrated that elevated NNMT levels could potentially exacerbate the production of the extracellular matrix and promote TIF by increasing p53 activity *via* a reduction in its DNA methylation.

## MATERIALS AND METHODS

### Animal model

Eight-week old male C57BL6J mice were purchased from Vitong Lihua (Beijing). Mice were raised in specific pathogen-free (SPF) conditions and housed with *ad libitum* access to food and water. Mice were randomly divided into four groups and subjected to either sham operation or unilateral ureteral obstruction (UUO) operation. To investigate the role of NNMT in renal fibrosis, mice were administered 0.25 mg/g/d NAM (Sigma-Aldrich, St. Louis, MO, USA) *via* intraperitoneal injection, starting 3 days before surgery and continuing daily for 7 days post-surgery. The renal fibrosis model was established as described previously (*Zhen et al., 2021*). The data presented in this manuscript are based on a separate mouse study, repeated under the same conditions as used before. For the time course study of NNMT, animals were sacrificed on days 3, 7, 10 and 14 after UUO. Other groups of mice were euthanized on day 7 post-surgery. The number of mice in each group was five to six and fed in animal cages, and a total of about 42 mice were used. Sample sizes is based on our previous experiments. The method of euthanasia was caused by cervical spine dislocation after intraperitoneal injection of an overdose of 1.5% pentobarbital sodium anesthetic (100 g/ml(w/v)). One of the following condition was used as benevolent endpoint and euthanized animals prior to the planned end of the experiment: animals lose 15–20% of their body weight rapidly, persistent hypothermia, obvious near-death manifestations. Mice were excluded from statistical analysis, that were euthanized prematurely or had large individual differences. All animal experiments were approved by the Ethics Committee for Animal Experiments at the Third Affiliated Hospital of Sun Yat-Sen University.

### Cell culture and treatment

The rat renal interstitial fibroblasts (NRK-49F) cell line was obtained from ATCC. NRK-49F cells were cultured in DMEM/F12 medium supplemented with 10% fetal bovine serum (Gibco/Life Technologies, New York, NY, USA). To assess the effect of NNMT on the TGF-β1-associated fibrosis, NRK-49F cells were starved for 12 h and then exposed to 1, 2, 5,10 or 20 ng/mL TGF-β1 (R&D Systems, Minneapolis, MN, USA) for 24 h. To transfect NNMT siRNA into NRK-49F cells, cells at 60–70% confluence were transfected with NNMT or scrambled siRNA using Lipofectamine 2000 for 24 h before stimulation with 10 ng/mL TGF-β1 for an additional 24 h. To investigate the role of NNMT *in vitro*, different concetration of NAM (1, 5, 10, or 20 mM) were applied for 1 h before stimulation with TGF-β1 (10 ng/mL) for 24 h.

### RNA extraction and quantitative real-time PCR

Total RNA was isolated from NRK-49F cells using TRIzol reagent and reverse transcribed according to the manufacturer's instructions (Vazyme, Nanjing, Jiangsu, China). Real-time PCR was performed to detect the expression of NNMT, CTGF, and GAPDH using SYBR Green Mastermix (Vazyme, Nanjing, Jiangsu, China). The primer sequences used in the experiments were as follows: (1) NNMT (rat) forward 5′-GAATCAGGCTTCA CCTCCAA-3′; reverse 5′-TCACACCGTCTAGGCAGAAT-3′; (2) GAPDH (rat) forward

5′-ACCATCTTCCAGGAGCGAGA-3′; reverse 5′-CTCGTGGTTCACACCCATCA-3′; (3) CTGF (rat) forward 5′-CTGACCTAGAGGAAAACATT-3′; reverse 5′-AGAAAGCTC AAACTTGACAG-3′. (4) NNMT (mice) forward 5′-TGTGCAGAAAACGAGATCCTC -3′; reverse 5′-AGTTCTCCTTTTACAGCACCCA-3′; (5) GAPDH (mice) forward 5′-AC TCCACTACGGCAAATTC-3′; reverse 5′-TCTCCATGGTGGTGAAGACA-3′.

## Western blot analysis

Cells and kidney tissues were homogenized and electrophoresed as previously described in *Li et al. (2019)*. Specifically, samples were electrotransferred to nitrocellulose membranesafter electrophoresis. The membranes were immunoblotted with primary antibodies, followed by horseradish peroxidase-conjugated or fluorescent-labeled secondary antibodies. An enhanced chemiluminescence detection system or Odyssey scanning was used to visualize blots. Primary antibodies such as anti-NNMT, anti-fibronectin, anti-α-SMA, anti-collagen I, anti-tenascin C, anti-p-p53, anti-p53 and anti-GAPDH used in this experiment were consistent with those described in *Zhen et al. (2021)*. Anti-pan-methylation was purchased from Abcam (Cambridge, UK).

## Immunoprecipitation

An immunoprecipitation kit was used to extract total protein from kidney tissues following the standard procedure (*Shu et al., 2022*). Protein complexes were obtained by incubating tissue lysates with either p53 antibodies or normal IgG overnight at 4 °C. Protein A/G agarose beads were added to the protein lysis for 4 h. Immunoprecipitates were washed in wash buffer for three times before dissolved in buffer containing sodium dodecyl sulfate (Sigma, St. Louis, MO, USA).

## Histological and immunohistochemical analyses

Masson-trichrome staining was carried out according to the manufacturer's protocol using 2 µm paraffin-embedded kidney sections. At least 10 random high-power fields (×400) in the cortex were evaluated to quantitatively assess renal fibrosis. The area was measured using Image Pro-Plus Software, and the average staining positive area (%) was calculated for each microscopic field.

## NNMT activation measurement

NNMT activity assay was based on previous studies (*Rudolphi et al., 2018*) which determined that the NNMT product reacts with ketones to form a fluorescent compound under alkaline conditions. The reaction solution was prepared by combining two parts of 1 mol/L nicotinamide and one part of 100 mmol/L SAM (Sigma-Aldrich, St. Louis, MO, USA). The fluorescence solution was prepared by mixing one part of 2 mol potassium hydroxide and 20% acetophenone in ethanol. A total of 97 µL of kidney tissue homogenate was blended with 3 µL substrate solution in a 96-well plate. Blank samples were prepared without substrate solution. The plate was incubated for 60 min at 37 °C. A series of 1-MNA dilutions (starting from 50 µmol/L) were prepared for the standard curve. After 60 min, 30 µL from each reaction well were transferred to a new black plate pre-filled with the 1-MNA standard, 3 µL substrate solution were transferred to each no-reaction wells.

A total of 120 µL fluorescence solution were added to each well, and the plate was incubated away from light at room temperature. After 10 min, 150 µL formic acid were added to each well and incubated for an additional 30 min at room temperature. Fluorescence parameter was set with λex 375 nm and λem 430 nm for 0.5 s. NNMT activity (pmol/min/mg) was calculated by determining the absolute amount of 1-MNA generated in samples, then divided by the reaction time and the absolute amount of sample total protein.

## Methylation-specific PCR

MSP was used to evaluate the methylation status of p53 promoter region in kidney tissue. The detail methods were based on previous study (*Najafipour et al., 2021*). A total of 2 µL of converted DNA with methylated and unmethylated primers was amplified in 20 µL reaction mixture. The MSP condition was: 94 °C for 5 min, followed by 35 cycles at 94 °C for 30 s (denaturation), 53 °C for 30 s (annealing), and 72 °C for 30 s (extension). The methylated primers were Forward: 5′-GGGAACGAGTGTTTAAAGTTAAGC-3′ and Reverse: 5′-AAAAAAATACGAAAAACCTATCGAA-3′, the unmethylated primers were Forward: 5′-GGAATGAGTGTTTAAAGTTAAGTGT-3′ and Reverse: 5′-AAAAAAATACAAAAAACCTATCAAA-3′.

## Single-cell RNA-sequencing and data processing

Kidneys from sham and UUO mice were minced into 1-mm pieces on ice and incubated in dissociation buffer, as previously described (*Si et al., 2019*). Specifically, samples were added into HBSS containing 10% fetal bovine serum on ice for 10 min to stop reaction. The resulting suspension was centrifuged at 500 g for 5 min after being passed through 70-um cell strainer (Falcon). Red blood cell lysate was added to the pellet for 3 min, then centrifuged at 500 g for 5 min and resuspended in phosphate-buffered saline. A hemocytometer with trypan blue was used to count the single-cell the suspension that was produced. About 10,000 cells were loaded to each channel and then partitioned into Gel Beads in emulsion in the Chromium instrument using the Single-Cell 3′ Reagent Kit v3 according to the manufacturer's protocol (10× Genomics). Library preparation including reverse transcription, barcoding, cDNA amplification, and purification was performed according to Chromium 10× v3 protocols. Libraries were sequenced on the Illumina HiSeq4000 platform.

Raw sequencing data were processed using Cell Ranger Single Cell Software Suite (v3.0.2) (10× Genomics). The *Mus musculus* genome reference (mm10) was used for reads alignment in accordance with 10× Genomics recommendations and reads were mapped to the reference genome using STAR (Spliced Transcripts Alignment to a Reference) with default setting. The mapped reads with valid barcodes and unique molecular identifiers (UMIs) were used to generate the gene-cell matrix.

The cells were reduced in dimension by principal component analysis (PCA) and then clustered using Uniform Manifold Approximation and Projection for Dimension Reduction (UMAP) tool. Cells were classified by using markers as previously described. Downstream analysis was performed in the R statistical software 3.6.3 (*R Core Team, 2020*;

*Feng et al., 2022*). Quality control (QC) filters were performed using the parameters which has been reported (*Park et al., 2018*): (1) cells with <200 genes were excluded; and (2) cells with >30% mitochondrial RNA reads were excluded. After QC filters, a total of 12,193 cells from two independent experiments were remained (5,093 cells from healthy and 7,100 cells from UUO mice).

## Statistical test

All data were expressed as means ± SD. Two groups' comparisons were analyzed using unpaired Student's t-test. For comparisons multiple groups were made using one-way ANOVA. The non-parametric Kruskal–Wallis test was used when data did not meet a normal distribution (SPSS software, version 19.0; SPSS, Inc., Chicago, IL, USA). 95% confidence interval, $p < 0.05$ was considered as statistically significant.

## RESULTS

### Increased NNMT expression in the fibrotic kidneys from mice with UUO

We initially investigated renal fibrosis and NNMT expression in the kidneys of mice subjected to UUO for 3, 7, 10, and 14 days. Western blot and real-time PCR results revealed that basal NNMT protein expression was low in the kidney cortices of mice. However, NNMT expression was significantly increased in the kidneys of UUO mice (Figs. 1A–1C), with its expression and renal fibrosis positively correlating with the duration of obstruction (Figs. 1D–1F).

### NNMT is primarily expressed in fibroblasts and positively correlates with fibroblast marker gene levels in the kidneys of UUO mice

To further investigate the role of NNMT in renal fibrosis, we conducted single-cell RNA sequencing (scRNA-seq) on UUO mouse kidneys. Figure 2A shows a uniform manifold approximation and projection (UMAP) of unsupervised clusters, and Fig. 2B presents markers for cell classification. An increased infiltration of fibroblasts and immune cells, consistent with UUO characteristics, was observed (Fig. 2C). Notably, scRNA-seq analysis of UUO-7d mouse kidneys revealed a marked increase and specific enrichment of NNMT expression in fibroblasts (Fig. 2D). The scRNA-seq data further supported the Western blot results, indicating a positive correlation between NNMT and fibroblast markers, as well as genes including α-SMA, fibronectin, and collagen I (Figs. 2E–2G). These findings suggest that the increased NNMT expression in the kidney after UUO injury may play a role in the development of renal fibrosis.

### NNMT mediates TGF-β1-induced renal fibroblast activation in NRK-49F cells

We further investigated the potential role of NNMT in mediating fibroblast activation *in vitro*. TGF-β1, as a master fibrogenic factor, was used to stimulate injury conditions in NRK-49F cells. Consistent with *in vivo* experiments, western blot and real time-PCR analysis indicated that the expression of NNMT was dependent on TGF-β1 concentration (Figs. 3A–3C). Western blot analysis of α-SMA, fibronectin (FN), and collagen I

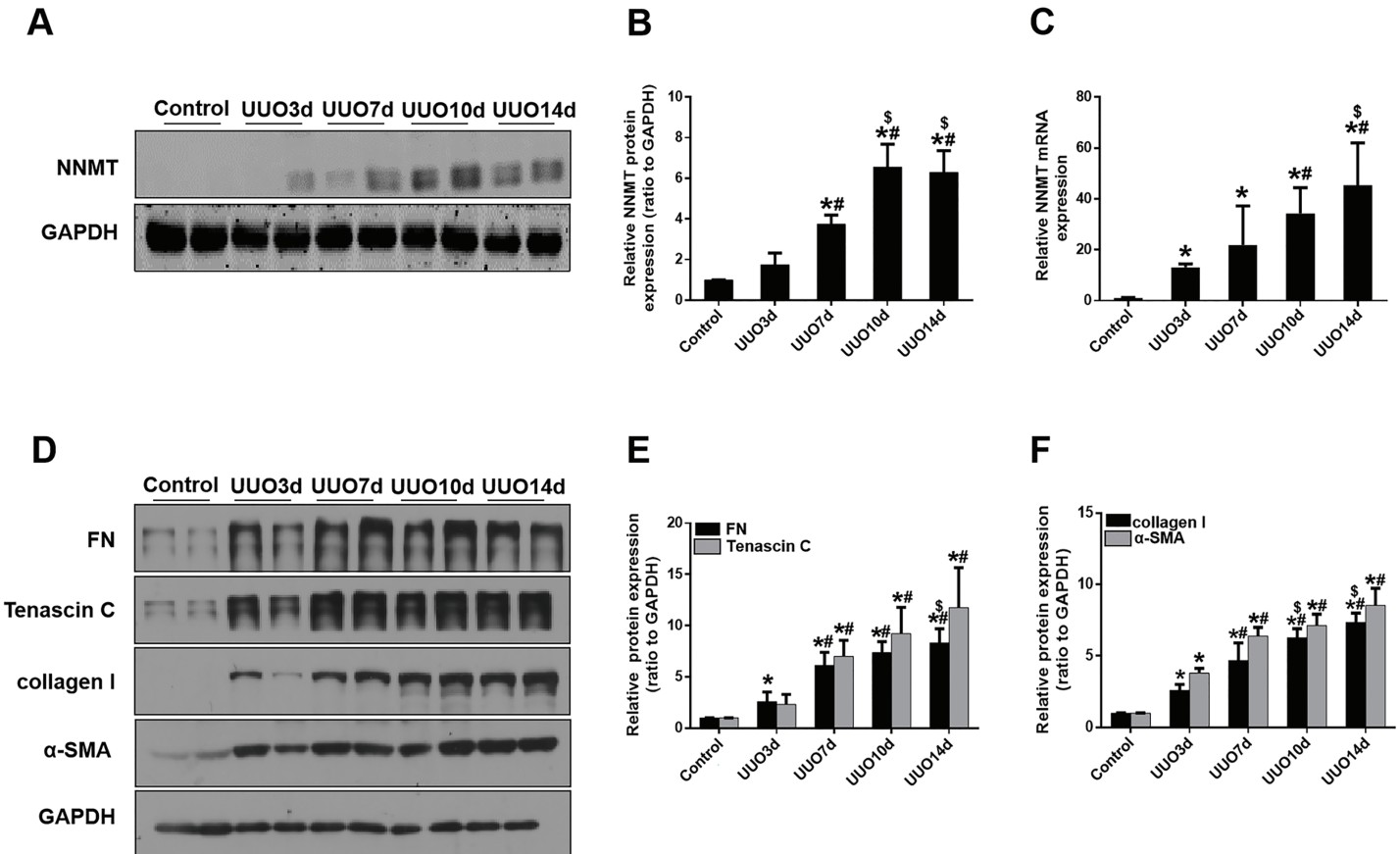

**Figure 1 The expression of NNMT is increased in mice UUO model.** Mice received UUO operation and were sacrificed at day 3, 7, 10, 14 respectively. (A) The expression of NNMT in UUO model at different times. (B) Graphic representation of relative protein level of NNMT normalized against GAPDH ($n = 5$). (C) Relative mRNA expression of NNMT in kidney at different times of UUO ($n = 5$). (D) Kidney expression of FN, Tenascin C, collagen I and α-SMA from control and UUO mice at different times, assayed by western blot. (E and F) Graphic representation of relative protein levels were normalized against GAPDH ($n = 5$). Data represent the mean ± SD. *$p < 0.05$ *vs.* control group; #$p < 0.05$ *vs.* UUO3d group; $$p < 0.05$ *vs.* UUO7d group.

demonstrated that matrix deposition was dependent on TGF-β1 concentration (Figs. 3D and 3E). To verify the direct role of NNMT in the activation of renal interstitial fibroblasts and matrix proteins production, NNMT siRNA was transfected either with or without TGF-β1 in NRK-49F cells. Figure 3F showed the transfection efficiency of NNMT in NRK-49F cells. The NNMT siRNA treatment notably attenuated TGF-β1-induced expression of α-SMA and Tenascin C (Fig. 3G), and confirmed by gray-level analysis (Fig. 3H). Additionally, TGF-β1-induced CTGF level were reduced by NNMT interference (Fig. 3I). These data suggest that NNMT is an important mediator of renal fibroblast activation.

## Nicotinamide supplementation inhibits NNMT activity and attenuates renal fibrosis in UUO mice

Previous studies have suggested that nicotinamide analogs can inhibit NNMT (*Ruf et al., 2018*), and our previous study demonstrated that nicotinamide supplementation can decrease NNMT expression (*Zhen et al., 2021*). To investigate the potential of nicotinamide in slowing CKD progression, mice were treated with daily intraperitoneal

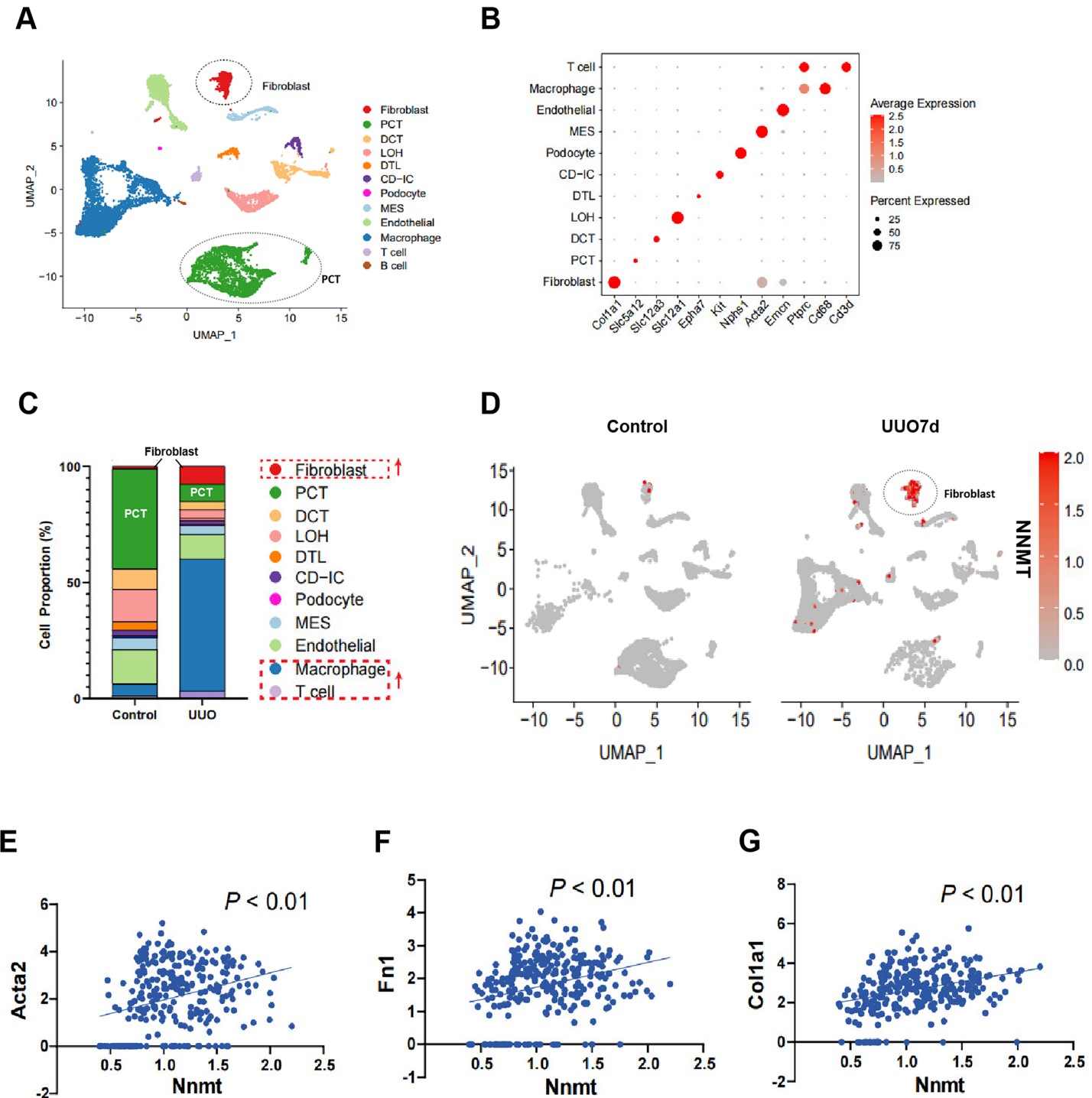

**Figure 2 scRNA-seq analysis reveals that NNMT is highly expressed in renal fibroblasts.** Single-cell RNA sequencing (scRNA-seq) is used to analyze normal and UUO-7d mice kidney. (A) Uniform manifold approximation and projection (UMAP) plot shows all single cells of normal and UUO mice kidney. (B) Marker gene for cell classification. (C) Distribution plot shows the cells proportion of normal and UUO mice kidney. (D) UMAP plot shows NNMT expression in different cell subsets of UUO. (E–G) scRNA-seq analysis the correlation between NNMT and fibrosis-related genes.

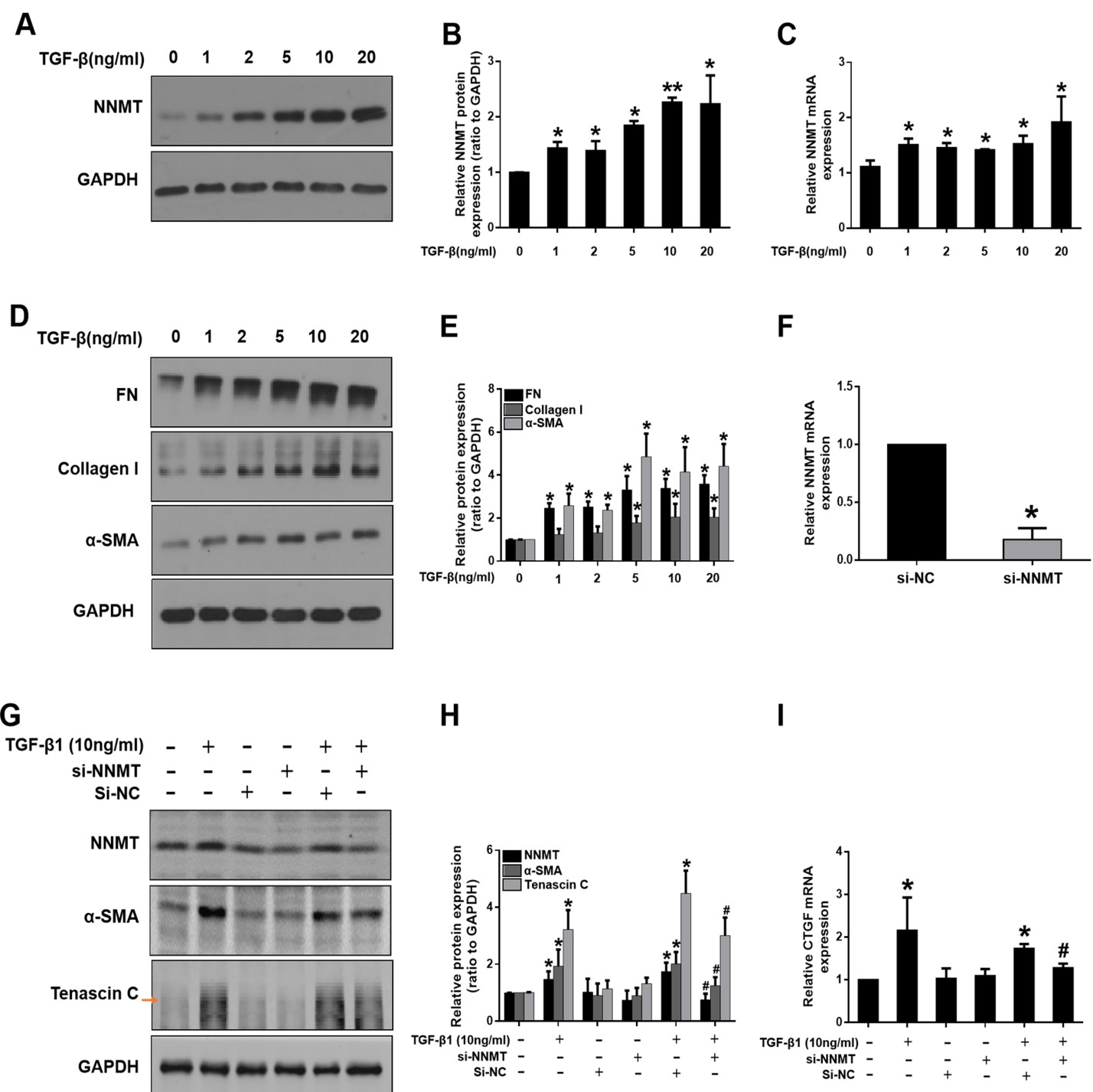

**Figure 3 The expression of NNMT is increased in TGF-β1 induced NRK-49F cells, and knockdown of NNMT abolishing the pro-fibrotic effect of TGF-β1.** (A) NNMT expression in NRK-49F after treatment with different concentrations of TGF-β1. (B) Graphic representation of relative protein level of NNMT normalized against GAPDH ($n = 3$). (C) Relative mRNA expression of NNMT in NRK-49F at different concentrations of TGF-β1. (D) Representative western blot of FN, collagen I and α-SMA. (E) Quantitative analysis of Fig. 3D ($n = 3$). *$p < 0.05$ *vs.* control group; **$p < 0.01$ *vs.* control group. (F) The mRNA level of NNMT in NRK-49F cells after treatment with si-RNA against NNMT or negative control. (G) NNMT, α-SMA and Tenascin C expressions in NRK-49F after treatment with si-RNA against NNMT or negative control. (H) Quantitative analysis of panel G ($n = 3$). (I) Relative mRNA expression of CTGF in NRK-49F after treatment with NNMT si-RNA ($n = 3$). Data represent the means ± SD. *$p < 0.05$ *vs.* control group; #$p < 0.05$ *vs.* TGF-β1+si-negative control (si-NC) group.

injections of nicotinamide for 3 days prior to UUO surgery. Masson and Sirius Red staining revealed that interstitial collagen fiber deposition was reduced following nicotinamide administration (Figs. 4A–4C). As NNMT's effectiveness relies on its catalytic activity, we also assessed the levels of 1-MNA in UUO-7d mouse kidneys to verify the effect of nicotinamide on NNMT activity. Our results showed that nicotinamide supplementation weakened NNMT activity (Fig. 4D) in the UUO mice model. Our previous study (Zhen et al., 2021) also demonstrated that mRNA expression of NNMT was lessened by nicotinamide supplementation (Fig. 4E). These findings suggest that nicotinamide replenishment can exert anti-fibrotic and anti-NNMT effects in UUO mice.

## Impairing NNMT using nicotinamide inhibits activation of renal interstitial fibroblasts *in vivo* and *in vitro*

Because the activation of renal interstitial fibroblasts is a key factor in the progression of renal fibrosis, we further investigated the effect of nicotinamide on interstitial fibroblast activation in the UUO model and NRK-49F cells, with a focus on NNMT. Immunoblot analysis revealed that the administration of nicotinamide significantly reduced NNMT expression and also decreased the levels of fibrotic markers, including α-SMA, fibronectin, and collagen I, in obstructed kidneys (Figs. 5A and 5C). To further confirm that NNMT regulates the activation of renal interstitial fibroblasts, we examined the impact of nicotinamide on renal fibroblast activation *in vitro*. We found that nicotinamide treatment inhibited the TGF-β1-induced increase in NNMT expression and activation of NRK-49F cells (Figs. 5B and 5D). These results suggest that NNMT activation is necessary for promoting renal fibroblast activation, which can be suppressed by nicotinamide treatment.

## Nicotinamide inhibits p53 signaling activation in the kidney following UUO injury

To further elucidate the mechanism by which NNMT contributes to pro-fibrotic responses, we utilized the STRING database to identify potential proteins that interact with NNMT. The results indicated a protein-protein interaction between NNMT and p53 (as shown in Fig. 6A). Previous research has demonstrated that nicotinamide supplementation can reduce renal fibrosis by activating the TGF-β/Smad3 signaling pathway (Zhen et al., 2021). Given that p53 can be activated by TGF-β1 and is also thought to be a co-factor in TGF-β1-mediated pro-fibrotic gene transcription, it is possible that NNMT may mediate p53 activation. To test this hypothesis, we assessed the effect of nicotinamide on p53 phosphorylation in the kidney after UUO injury. We found that UUO injury led to increased p53 phosphorylation, which was suppressed by nicotinamide treatment (as demonstrated in Figs. 6B and 6C). Previous studies have shown that NNMT catalyzes N-methylation of pyridine-containing compounds using SAM as a methyl donor, thereby regulating cellular methylation potential to impact various epigenetic processes (van Haren et al., 2016; Babault et al., 2018). Based on this, the researchers investigated whether NNMT could decrease p53 methylation to regulate renal fibrosis. Both immunoprecipitation (as shown in Figs. 6D and 6E) and methylation-specific PCR (as demonstrated in Fig. 6F) indicated that both the p53 protein's pan-methylation and DNA

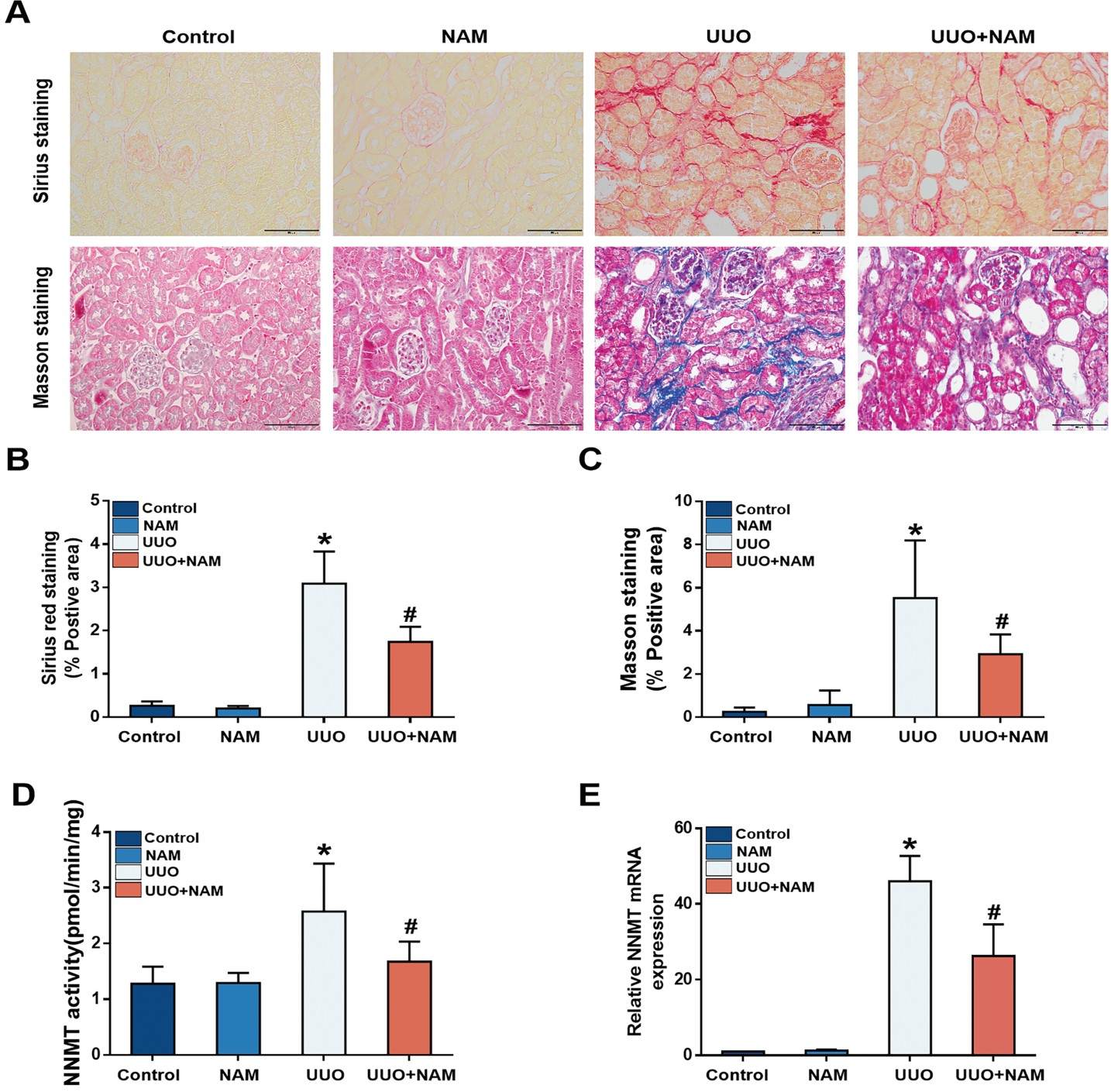

**Figure 4 NAM as an inhibitor of NNMT and the anti-fibrotic effects of NAM in UUO model.** Mice received NAM (0.25 mg/g) 3 days before UUO operation, and all were killed at day 7 after UUO. (A) Representative photomicrographs of sirius red and masson staining from kidneys of control, UUO and NAM-treated UUO mice (magnification, ×400). (B and C) Renal interstitial fibrosis scores based on sirius red or masson staining. (D) The activity of NNMT was decreased *via* injecting NAM. (E) The relative mRNA expression of NNMT in kidney cortexes of UUO7d model (n = 5). Data are expressed as mean ± SD. *p < 0.05 *vs.* control group; #p < 0.05 *vs.* UUO group.

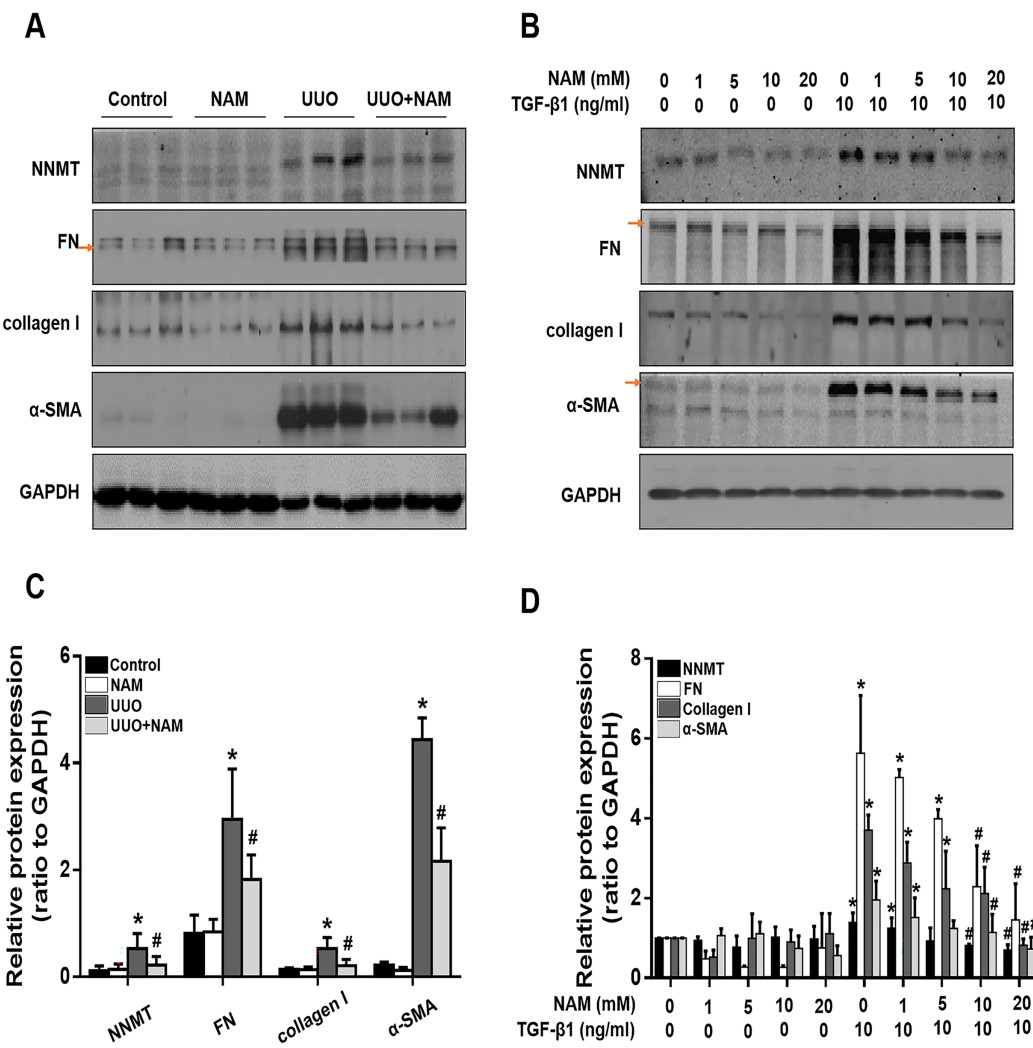

**Figure 5 Supplementary with NAM decreases the fibrosis *in vivo* and *in vitro*.** (A) Represent expression of NNMT, FN, collagen I and α-SMA from control, UUO and NAM-treated UUO mice kidneys, assayed by western blot. NRK-49F were pretreated with NAM (1, 5, 10, 20 mM) for 1 h and then coincubated with TGF-β1for 24 h. (B) Western blot analysis of NNMT, FN, collagen I and α-SMA. (C) Quantitative analysis of Fig. 5A ($n = 5$). $^*p < 0.05$ *vs.* control group; $^{\#}p < 0.05$ *vs.* UUO group. (D) Quantitative analysis of Fig. 5B ($n = 3$). $^*p < 0.05$ *vs.* control group; $^{\#}p < 0.05$ *vs.* TGF-β1 group. All data are expressed as mean ± SD. 

methylation levels were decreased following UUO injury, but this phenomenon was reversed by NNMT inhibition triggered by nicotinamide treatment. Collectively, these findings suggest that NNMT contributes to renal fibrosis by weakening methylation and increasing p53 activation.

## DISCUSSION

Previous research has indicated that overexpression of NNMT can have negative impacts in diet-induced obesity (*Trammell & Brenner, 2015*; *Brachs et al., 2019*). Additionally, high stromal expression of NNMT—as a master metabolic regulator—has been correlated with poor prognosis in various types of human cancers (*Eckert et al., 2019*). However, little is

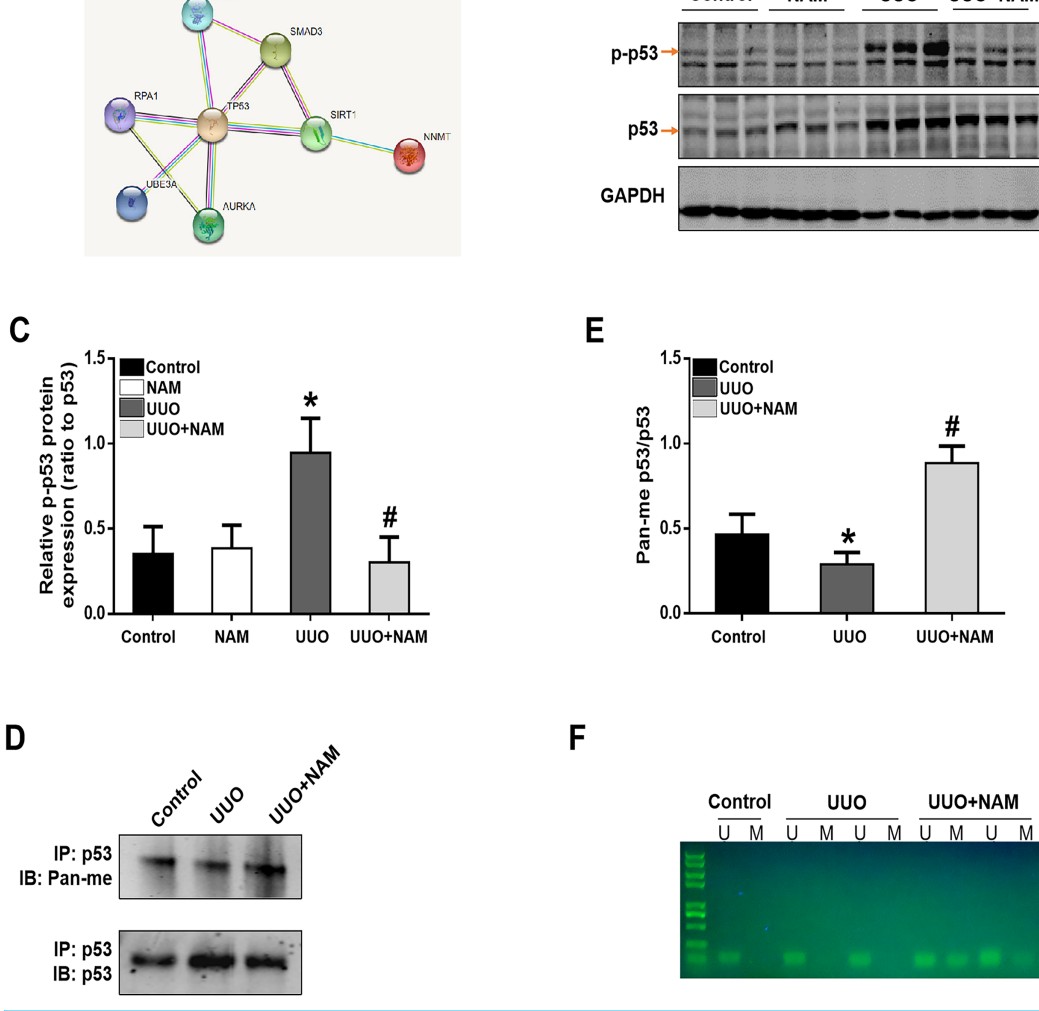

**Figure 6 NAM ameliorates renal fibrosis *via* increasing the methylation level of p53.** (A) STRING database showed the possible interaction between NNMT and p53. (B) Representative western blots show the levels of phosphorylated p53 (p-p53) in kidney cortexes of different groups. (C) Graphic representation of relative protein levels of p-p53 normalized against its total protein ($n = 6$). (D) Pan-methylated and total protein of p53 in kidney cortexes. (E) The relative protein expression of p53 methylation were normalized against total protein ($n = 4$). (F) MSP detected DNA CpG island methylation of p53 (U, non-methylated primer amplification product; M, methylated primer amplification product). Data are expressed as mean ± SD. *$p < 0.05$ *vs.* control group; #$p < 0.05$ *vs.* UUO group.

currently known regarding the role of NNMT in renal fibrosis. To investigate the impact of NNMT on kidney fibrosis, the researchers generated TGF-β1-treated NRK-49F cells and UUO-induced mouse models of renal fibrosis. Our findings indicate that high levels of NNMT accumulate in UUO-induced TIF and the activation of renal interstitial fibroblasts. Furthermore, the data demonstrates that NNMT decreases the methylation level of p53 and increases the expression of CTGF, contributing to the pro-fibrotic role of NNMT. Conversely, inhibition of NNMT may prevent the progression of renal fibrosis. These

results suggest that NNMT plays a critical pro-fibrotic role and could be a promising therapeutic target in cases of chronic fibrotic kidney disease.

NNMT is an intracellular methyltransferase that catalyzes the formation of methylnicotinamide (MNAM) from nicotinamide (NAM) by using the methyl group of S-adenosylmethionine (SAM). NNMT is primarily expressed in the liver, adipose tissue, kidney, and heart of mammals (*Riederer et al., 2009*). SAM serves as a universal methyl donor for histones, non-histones, DNA, RNA, and other metabolites. NNMT can reduce DNA, RNA, histone, and protein methylation by depleting SAM and participating in various biological processes, making it a novel metabolic regulator (*Roberti, Fernández & Fraga, 2021*). NNMT is highly expressed in fibroblasts of multiple tumors in various tissue types, promoting the expression of cancer-associated fibroblast markers and collagen contractility by reducing the methylation of gene promoter regions and histones, ultimately accelerating tumor growth, progression, and metastasis (*Sun et al., 2022*). Given the similarities between fibroblasts, this study aimed to investigate the role of NNMT in renal fibroblasts. Both *in vivo* and *in vitro* findings revealed that NNMT expression was positively correlated with the degree of renal fibrosis. scRNA-seq analysis of UUO mouse kidneys confirmed that NNMT was significantly elevated and predominantly expressed in renal fibroblasts. The results also showed that NNMT positively correlated with a set of fibroblast activation signature genes. *In vitro*, knocking down NNMT inhibited TGF-β1-induced proliferative activation of NRK-49F cells and decreased the expression of the pro-fibrotic gene CTGF, suggesting that chronic renal injury leads to NNMT accumulation, promoting fibroblast activation.

Nicotinamide (NAM), a product of vitamin B3, generates $NAD^+$ *via* a salvage synthetic pathway and is required to regulate energy metabolism and cellular lifespan (*Tran et al., 2016*). NAM and its related analogs have been shown to inhibit the activity of NNMT. In this study, we used NAM as a mimic NNMT inhibitor to further investigate the role of NNMT in animal models. Supplementation with NAM inhibited the expression and activity of NNMT, consistent with previous findings. Our prior research found that supplementing with NAM ameliorated renal fibrosis by increasing renal $NAD^+$ content and its downstream sirtuin1 activity (*Zhen et al., 2021*). NNMT inhibition did increase the amount of NAD+, which may have been due to a decrease in NNMT's consumption of NAM. Additionally, after NAM supplementation, the increased substrate of NNMT may lead to the production of 1-methylnicotinamide (1-MNA), which may have the ability to negatively regulate NNMT. While we have established a dose-dependent correlation between NNMT and NAM, it is unclear whether NAM could be act as a direct inhibitor of NNMT due to its broad range of action targets. Additionally, the UUO group is the only one to experience a decrease in NNMT activity and expression in response to NAM supplementation; the control group did not. This possibly indicates that the NNMT expression is low in healthy mouse kidneys, as a result that there is no appreciable difference between control and UUO groups in how NAM supplementation affects NNMT activity and expression.

The classical tumor suppressor p53 is activated in response to cellular DNA damage, oxidative stress, and other factors (*Chibaya et al., 2021*). In recent years, p53 has been

identified as a key regulator in the progression of renal fibrosis. P53 is thought to be a key co-factor in TGF-β1-mediated pro-fibrotic gene transcription, working with TGF-β1 to regulate cell growth inhibition and extracellular matrix remodeling. Various post-translational modifications, such as phosphorylation, acetylation, and ubiquitination, influence p53 transcriptional activity. Activated p53 is recruited to target promoter region to promote gene transcription and is involved in the regulation of apoptosis, cell cycle arrest and other cell biological processes. CpG islands are CG-rich DNA regions located upstream of genes and consist of gene promoters, that regulate gene expression *via* varying degrees of methylation. Methylation of CpG islands in the p53 promoter region has been shown to reduce p53 expression and prevent it from binding to transcription factors. Furthermore, p53 is also considered as one of the few non-histone proteins regulated by lysine methylation (*Scoumanne & Chen, 2008*). The p53 protein has twenty lysines, six of which are located in p53 binding domain (p53-BD). Three of these six lysines are specifically methylated by histone lysine methyltransferases, KMT5 (Set9), KMT3C (Smyd2) and KMT5A (Set8). KMT5-induced methylation of p53-K372 increases the transcriptional activity of p53 on target genes (*Allis et al., 2007*). However, methylation of p53-K370 by KMT3C and monomethylation of p53-K382 by KMT5A inhibit the transcriptional activation of p53 on p21, MDM2, and PUMA, resulting in p53-mediated cell cycle arrest and reduced cell apoptosis (*Shi et al., 2007*). Additionally, arginine residues of p53 can also be methylated. Methylation of p53-R333, R335 and R337 by protein arginine methyltransferase 5 (PRMT5) could regulate p53 response to DNA damage and affect p53 target gene specificity (*Jansson et al., 2008*). In this study, we found an interaction between NNMT and p53 using a protein-protein interaction network. In UUO mouse model, inhibiting NNMT ameliorated renal fibrosis and increased the levels of p53 protein pan-methylation and its DNA methylation, which might be mainly due to the fact that supplementation with NAM reduced the methyl groups consumption of SAM by NNMT. However, whether NAM or NNMT interacts with p53 methyltransferase, as well as the specific sites of increased p53 methylation following NAM supplementation, were not thoroughly investigate in this study. In combination with our previous study, we suggested that NAM supplementation reduced p53 phosphorylation and acetylation. However, whether there are post-translational interaction between p53 methylation and its phosphorylation and acetylation has also not been investigated.

## CONCLUSION

In conclusion, our findings show that NNMT is highly expressed in renal fibroblasts and correlates with the degree of fibrosis after kidney injury. TGF-β1-induced myofibroblast activation was significantly reduced *in vitro* after NNMT knockdown. *In vivo*, inhibiting NNMT activity increased p53 protein and DNA methylation, which inhibited the expression of the downstream pro-fibrotic target factor CTGF, weakening proliferation and activation of fibroblasts. Overexpression of NNMT reduces $NAD^+$ synthesis by depleting NAM, and it also decreases methylation activity of pro-fibrotic genes involved in renal fibrosis by depleting the methyl group of S-adenosylmethionine. Our findings

suggest that targeting NNMT as an useful control strategy for progressive renal fibrosis could be beneficial.

### Funding

This work was supported by grants from "three big" construction big scientific research projects of Sun Yat-sen University (82000-18843406) to Hui Peng. The funders had no role in study design, data collection and analysis, decision to publish, or preparation of the manuscript.

### Grant Disclosures

The following grant information was disclosed by the authors:
Sun Yat-sen University: 82000-18843406.

### Competing Interests

All authors declare that they have no competing interests.

### Author Contributions

- Xin Zhen conceived and designed the experiments, performed the experiments, analyzed the data, prepared figures and/or tables, authored or reviewed drafts of the article, and approved the final draft.
- Yuxiang Sun analyzed the data, prepared figures and/or tables, and approved the final draft.
- Hongchun Lin performed the experiments, authored or reviewed drafts of the article, and approved the final draft.
- Yuebo Huang performed the experiments, prepared figures and/or tables, and approved the final draft.
- Tianwei Liu performed the experiments, prepared figures and/or tables, and approved the final draft.
- Yuanqing Li analyzed the data, prepared figures and/or tables, and approved the final draft.
- Hui Peng conceived and designed the experiments, authored or reviewed drafts of the article, and approved the final draft.

### Data Availability

The raw data of western blots are available in the Supplemental File.

### Supplemental Information

Supplemental information for this article can be found online at http://dx.doi.org/10.7717/peerj.16301#supplemental-information.

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
