# Peer review of "Elucidating the role of nicotinamide N-methyltransferase-p53 axis in the progression of chronic kidney disease"

_PeerJ, doi:10.7717/peerj.16301_

## Round 0.1 · original submission · Major Revisions

In addition to the reviewers' comments, please clearly highlight the novelty of the proposed manuscript and add relevant previously published papers in the field to the cited references.

Reviewer 1 ·

Basic reporting

The present study, performed by Zhen and colleagues, reveals a promising breakthrough in treating chronic kidney disease (CKD) and renal interstitial fibrosis (TIF). Nicotinamide N-methyltransferase (NNMT) has been found to increase in kidneys with TIF, and its role in promoting fibroblast activation and renal fibrosis has been established. By inhibiting NNMT, the researchers observed a reduction in fibroblast activation and an increase in DNA methylation of p53, leading to a decrease in its phosphorylation. These findings suggest that targeting NNMT could be a novel approach to preventing and treating renal fibrosis. It offers hope for more effective therapies to halt the progression of CKD and related conditions. This study provides valuable insights into the mechanisms underlying renal fibrosis and opens up new possibilities for future research and treatment development.

Overall, the manuscript has been written very well. Please check the minor typos throughout the manuscript. For example, line 348 “inhibite”.

Experimental design

1. Please provide NNMT knockdown efficiency validation for Figures 3F-3I.

2. The RT-qPCR primer sequences for mouse genes are missing.

Validity of the findings

When two levels are tested (e.g., TGF-b, vehicle, and siNC, siNNMT), a two-way ANOVA should be used, or a non-parametric equivalent (such as Kruskal-Wallis test), if the assumptions of parametric testing are not met. This applies to Figures 3G-3I, 4B-4E, and 6D.

Reviewer 2 ·

Basic reporting

The manuscript has been written clearly. Overall the relevant literature has been cited, although the reference to support the statement made in lines 278-281 should include Van Haren et al, 2016 [doi 10.1021/acs.biochem.6b00733] in which is demonstrated that NNMT can methylate different pyridinyl compounds and show that NAM does not inhibit NNMT.
The structure of the article reads well and the raw data has been shared. It was noted that several figures and bar graphs published in Zhen et al, 2021 [DOI: 10.1159/000510943] seemed to be reused in this manuscript. While the current manuscript does contain sufficient novel data, the author is requested to describe the overlap and difference between the previous study and the current manuscript to comprehend the added value of the current manuscript.
The abbreviations RIF and TIF are both used. From the manuscript it seems RIF stands for Renal Interstitial Fibrosis and TIF stand for Tubulointerstitial fibrosis. Please correct the abbreviations used throughout the manuscript where applicable.
Please amend the following:
Line 98-99: sentence needs to be corrected
Line 106: change exposure to exposed

Experimental design

The authors described the role of NNMT in renal fibrosis with a focus on the effect of NNMT on p53. While no direct link between NNMT and p53 could be established, the effect of UUO on p-p53, me-p53 and NNMT seem to be linked.
The author is requested to comment on the fact that no methylation of p53 could be observed by MSP in the control group in figure 6F, nor in the UUO group, but only in the UUO+NAM group.
In addition, the raw data corresponding to the data presented in figure 6D seems to be different. Please comment on the potential effect of this difference on the bar graphs and results discussed.

Validity of the findings

Nicotinamide is not known to inhibit NNMT in the same way nicotinamide analogs can. Please provide the reference to your previous study as stated in line 245 (and in other sections of the manuscript). Unless a direct dose-dependent inhibition of NNMT by NAM can be demonstrated, please amend the statement of NAM being an NNMT inhibitor and discuss the possibility that increased nicotinamide levels can lead to increased metabolites, either 1-methylnicotinamide, acting as a feedback inhibitor of NNMT, or downstream nicotinamide metabolites such as NAD+ improving the UUO phenotype.

Reviewer 3 ·

Basic reporting

Presented manuscript by Zhen X. et al. lacks the novelty because similar results are already published by same authors - doi: 10.1159/000510943. However, authors added some new information about NNMT and p53 axis but lacks mechanistic explanation for p53 mediated functions in CKD. This manuscript in current form does not add any significant information in already existing knowledge.

Experimental design

Here, majority of results claim NNMT expression and its importance during UUO CKD mice model. However, effect of NNMT and NAM supplementation has already been extensively studied on kidney disease by following recent articles - DOI: 10.1096/fj.202100913RRR and DOI: 10.1038/s41598-022-10476-6.

Validity of the findings

No comment

---

## Round 0.2 · Minor Revisions

Please clarify in more detail the methodology.

Reviewer 1 ·

Basic reporting

The authors have addressed all my concerns, and I have no further comments.

Experimental design

None.

Validity of the findings

None.

Additional comments

None.

Reviewer 2 ·

Basic reporting

Overall, the adjustments made resolve the comments made.
The authors decided to use TIF as the common abbreviation for kidney fibrosis. While this is acceptable, renal interstitial fibrosis (TIF) should be changed to tubulointerstitial fibrosis (TIF) in line 41 to avoid confusion.

Experimental design

It remains unclear if the data form the mouse studies presented by Zhen et al in 2021 have been (partially) reused or whether a similar mouse study with the same design has been performed. From the bar graphs and figures presented it seems that at least some of the data has been reused. If so, clear reference to the previous paper should be made in those cases. If not, please make clear that the data presented are based on a separate mouse study, repeated under the same conditions as used before..

Validity of the findings

While the questions were generally answered satisfactorily, very limited adjustments to the text of the manuscript have been made.
The authors keep the statement that nicotinamide is an inhibitor of NNMT, while only indirect effects of NAM on NNMT are shown. Previous studies have shown that NAM does not affect NNMT activity up to concentrations of up to 50 mM and that NAM supplementation improves TIF through boosting of sirtuins rather than inhibition of NNMT. NAM supplementation is more likely mimicking NNMT inhibition, as inhibition of NNMT would result in an increase in NAM as well. In addition, the reduction of NNMT activity and expression upon NAM supplementation is only seen in the UUO group, but not in the control group.
As stated in the response document, this discussion should be reflected and emphasized more clearly in the manuscript. Alternatively, dose-dependent inhibition NNMT by NAM providing an IC50 value should be demonstrated.

---

## Round 0.3 · accepted · Accept

No further comments.

Reviewer 2 ·

Basic reporting

no comment

Experimental design

no comment

Validity of the findings

no comment

Additional comments

All points have been resolved adequately